# Bacilli Rhizobacteria as Biostimulants of Growth and Production of Sesame Cultivars under Water Deficit

**DOI:** 10.3390/plants12061337

**Published:** 2023-03-16

**Authors:** Giliard Bruno Primo de Lima, Erika Fernandes Gomes, Geisenilma Maria Gonçalves da Rocha, Francisco de Assis Silva, Pedro Dantas Fernandes, Alexandre Paulo Machado, Paulo Ivan Fernandes-Junior, Alberto Soares de Melo, Nair Helena Castro Arriel, Tarcisio Marcos de Souza Gondim, Liziane Maria de Lima

**Affiliations:** 1Agricultural Sciences, UEPB, Campina Grande 58429-500, PB, Brazil; 2Biological Sciences, UEPB, Campina Grande 58429-500, PB, Brazil; 3Embrapa Algodão, Campina Grande 58428-095, PB, Brazil; 4Agricultural Engineering, UFCG, Campina Grande 58429-900, PB, Brazil; 5Department of Basic Health Sciences, Federal University of Mato Grosso, Campo Grande 78060-900, MT, Brazil; 6Embrapa Semiárido, Petrolina 56302-970, PE, Brazil

**Keywords:** abiotic stress, water restriction, rhizobacteria, osmoprotection system, leaf gas exchange, sesame

## Abstract

A strategy using bacilli was adopted aiming to investigate the mitigation of the effects of water deficit in sesame. An experiment was carried out in a greenhouse with 2 sesame cultivars (BRS Seda and BRS Anahí) and 4 inoculants (pant001, ESA 13, ESA 402, and ESA 441). On the 30th day of the cycle, irrigation was suspended for eight days, and the plants were subjected to physiological analysis using an infrared gas analyzer (IRGA). On the 8th day of water suspension, leaves were collected for analysis: superoxide dismutase, catalase, ascorbate peroxidase, proline, nitrogen, chlorophyll, and carotenoids. At the end of the crop cycle, data on biomass and vegetative growth characteristics were collected. Data were submitted for variance analysis and comparison of means by the Tukey and Shapiro–Wilk tests. A positive effect of inoculants was observed for all characteristics evaluated, contributing to improvements in plant physiology, induction of biochemical responses, vegetative development, and productivity. ESA 13 established better interaction with the BRS Anahí cultivar and ESA 402 with BRS Seda, with an increase of 49% and 34%, respectively, for the mass of one thousand seeds. Thus, biological indicators are identified regarding the potential of inoculants for application in sesame cultivation.

## 1. Introduction

A broadly defined group of rhizobacteria has been associated with plants, as promoters of plant growth and root development by several mechanisms, such as biological nitrogen fixation [1,2], in the release and uptake of insoluble nutrients (e.g., iron and phosphorus), stimulation of phytohormone synthesis [3], suppression of pathogens through the production of antibiotics and siderophores [2,4], among others, establishing beneficial associations with their host plants.

In drylands (arid and semi-arid regions), the association of plants (both crops and native species) with stimulating rhizobacteria is important to the plant’s establishment and development, since in addition to providing nutrients and stimulating growth, these microbes also help plants cope with drought stress [5,6,7,8]. These bacteria could be used as inoculants in agriculture since biofertilizers with high concentrations of microorganisms act on the growth and development of plants, increasing crop production, preserving soil life, reducing production costs, without causing damage to water resources or emissions of pollutants, and acting upon soil bioremediation [9,10]. In addition, due to the mitigating action on the negative effects caused by biotic and abiotic stresses [11,12], inoculants can be used as a strategy to promote various agricultural crops in arid and semi-arid environments. In drylands, the effects of inoculants are already observed in several crops, for example, maize (*Zea mays*) [13], sorghum (*Sorghum bicolor*) [14], pear-millet (*Pennissetum glaucum*) [15], and sesame (*Sesamum indicum*) [16,17], indicating the potential of selected bacteria for agricultural applications in the field.

Sesame (*Sesamum indicum* L.) is an agricultural food crop that is oleaginous and has seeds with excellent quality oil as well as antioxidants from the presence of sesamol, sesamolin, and sesamin [18]. This crop is used in the production of various foods, in the medicinal and pharmaceutical industries, and also consumed in nature and in animal feed. It is easy to grow and tolerant to certain periods of drought, which makes it an ideal alternative for small and medium rural producers [19]. However, to achieve maximum yield, it requires well-distributed rainfall during the different phases of its cycle [20]. In this context, the application of stimulating bacteria could benefit plant growth under water deficit conditions.

Water deficit is limiting for water relations at the cellular level or for the whole organism of plants, causing economic losses in agriculture [21]. Rhizobacteria can act on the regulation of plant physiology and produce compounds that act as mediators for abiotic stress in plants by inducing the expression of specific genes [22,23,24]. Drought is a serious problem for agricultural crops, and to overcome this limitation, technologies are emerging, such as the use of new inoculants [5,7], aiming to promote increased production, especially in environments where climatic conditions are limiting, such as arid and semi-arid environments.

Based on this perspective, this study aimed to investigate the vegetative growth and yield of two sesame cultivars in interaction with *Bacillus* spp. inoculants, as well as the physiological and biochemical behavior of plants, under water deficit conditions.

## 2. Results and Discussion

### 2.1. Gas Exchange and Fluorescence

For both cultivars, Ci (internal concentration of CO_2_) increased in most treatments under water stress conditions (Figure 1a). Sesame genotypes under salt stress with average Ci below 200 μmol mol^−1^ indicate low photosynthetic activity [25]. Decreases in Ci present stomatal limitation, impairing photosynthetic performance, as the greater the stomatal opening, the greater the diffusion of carbon dioxide to the substomatic chamber, increasing Ci and consequently favoring photosynthesis. When comparing the data referring to Ci and iEC (instantaneous efficiency of carboxylation) (Figure 1e), it was possible to observe a low efficiency of carboxylation, implying that the accumulation of CO_2_ present, especially in treatments under water stress, was not transferred to the substomatal chamber where it would be used in the incorporation of photoassimilates resulting from the photosynthetic process.

As expected, with the decrease in the other variables, the A (photosynthesis) (Figure 1b) was also affected in plants under water restriction, with an average reduction of 60%. Photosynthesis decreases as a function of stomatal closure and the persistence of water deficit [26]. The reduction in photosynthetic rate is correlated with several factors, such as the amount of water absorbed, the CO_2_ fixed by the plant due to stomatal closure, as well as by the diffusive resistance of the stomata that limits the gaseous conduction of the leaf [27].

For E (transpiration), a reduction was observed in both cultivars in all treatments under water deficit (Figure 1c). The reduction in E comes from stomatal closure since plants under water restriction or at high temperatures tend to close their stomata so as not to lose water, as highlighted in studies with sesame plants under water deficit [28,29]. E is a crucial factor for processes such as leaf temperature regulation. E occurs in the stomata, also known as guard cells, which are microscopic pores present in the leaf epidermis that capture biotic and abiotic stimuli from the internal or external environment and respond quickly with mechanisms to close the stomata under unfavorable conditions, such as water deficit [30,31].

The gs (stomatal conductance) showed no significant difference in the triple interaction in the cultivars under water deficit; however, in the irrigated condition, ESA 441 improved the dynamics of stomatal conduction in both cultivars (Figure 1d). The interaction of cultivars with the water regime showed similar behavior for both cultivars in the stress condition. The interaction between the water regime and the treatments showed a significant difference; therefore, the photosynthetic metabolism had its functioning compromised by the low absorption of CO_2_ or the direct effect of the stress caused by water deficit [32].

The A/Ci ratio implies the instantaneous efficiency of carboxylation (iEC). iEC closely matches intracellular CO_2_ concentration and carbon dioxide assimilation rate. Variations in the optimal temperature (between 20 °C and 30 °C) or factors considered stressful to plants, such as salinity and water deficit, can cause a restriction in the flow of CO_2_ to the carboxylation site [33], thus hindering cells’ metabolism in the use of substrate for plant cell biosynthesis. For iEC in the triple interaction, inoculants and nitrogen management provided greater carboxylation efficiency in the irrigated condition (Figure 1e). No statistical difference was observed in the treatments in BRS Anahí in the stressed condition; however, BRS Seda was more favored by inoculation with pant001, ESA 13, and WN in the same condition, being relatively more efficient.

ESA 402 proved to be efficient in maintaining a higher iTE standard in both cultivars, especially under water restriction (Figure 1f). Similar results were described in sesame plants under water stress and with the application of salicylic acid [34]. The same bacteria also improved the water use efficiency of sorghum under drought [7], indicating the potential of this strain for inoculating multiple crops.

When the fluorescence (Fo, Fv, Fm, and Fv/Fm) was measured, it was possible to identify positive interactions within the adopted significance levels (Figure 2). Plants once subjected to abiotic stresses (salinity, water deficit, heat stress, among others) show changes in the state of membranes, implying changes in the thylakoids’ function in chloroplasts, triggering changes in the characteristics of fluorescence signals. Despite the consequences imposed by different types of stress, sesame presents good phenotypic plasticity [35]. 

### 2.2. Biomass, Growth Measures and Nitrogen

A high sensitivity to water deficit was observed in both cultivars in the first 24 h after watering was suspended, with signs of wilting, leaf curl, curving of the main stem, side branches, and the presence of trichomes. There was an accentuated floral abortion and leaf abscission in all stressed plants (Figure 3). This behavior is a strategy used by plants to save water and energy, given the critical moment to which they were subjected [36].

Sesame plants inoculated with the bacilli showed a positive interaction for plant height (Figure 4a). For the cultivar BRS Anahí there was no significant difference for the treatments in the irrigated condition; however, in the condition with water deficit pant001 and WN promoted the best averages for the variable plant height, without statistically differing from the strains ESA 13 and ESA 441. In the water deficit condition for BRS Seda, treatments with ESA 441, WN, and NN were the most significant for plant height (Figure 4a). In corn, rhizobacteria can potentially reduce fertilization with inorganic nitrogen without affecting growth parameters, proving the efficiency of *Bacillus* spp. as growth promoters when faced with a chemical nitrogen fertilizer [37]. Studies with sesame plants (BRS Seda) at different irrigation levels showed averages like those found in this work [38,39]. 

For the stem diameter variable, no significant difference was observed for the inoculants in the cultivar BRS Anahí, in the irrigated condition; in the water deficit condition, pant001 and nitrogen management presented the best means. The cultivar BRS Seda had a positive interaction in the irrigated condition with pant001; however, when submitted to water restriction, the strain ESA 402 was more efficient in the stem diameter, but it did not differ statically from the other treatments (Figure 4b).

When the shoot dry mass was evaluated for the cultivar BRS Anahí, in the irrigated condition there was a significant difference and the treatments inoculated with the bacilli showed the best means; in the water deficit condition, strain ESA 402 and WN presented the highest means, while pant001 and ESA 13 also showed positive effects when compared to NN (Figure 4c). In the cultivar BRS Seda inoculated with ESA 402, there was a significant increase in the dry mass of the shoots in the water deficit condition. Due to the characteristics of each cultivar, there was a significant difference in shoot dry mass between them (Figure 4c). Significant effects of water restriction on sesame were observed with a reduction in shoot dry mass, plant height, number of capsules per plant, and sesame productivity from the thirtieth day after planting [38]. ESA 13 already proved to be an efficient plant growth promoter for rice (*Oryza sativa*) [6], while ESA 402 showed positive effects on sorghum under full irrigation [40] and water deprivation conditions [7]. The results observed for the sesame genotypes in the present study agree with the potential of both bacilli to compose multi-crop inoculants for drylands.

In relation to dry root mass, in the irrigated condition, the cultivar BRS Anahí inoculated with ESA 13 presented the best average (Figure 4d); with water deficit, pant001 promoted the highest dry root mass, followed by ESA 13, WN, and ESA 402; with ESA 441, the dry mass of the root was reduced by more than 50% in relation to the irrigated condition. In the BRS Seda cultivar, the results for dry root mass were quite similar between irrigated and non-irrigated conditions. However, pant001 was the strain that provided the highest mean among the inoculated treatments (Figure 4d). In an experiment with peanut genotypes treated with bacterial strains and under water deficit, minimal differences were observed in relation to water restriction [41]. Despite the plants showing reductions in root growth, the bacterial strains were efficient in promoting growth and water deficit attenuation, emphasizing the importance of using inoculants to mitigate the effects of water deficit [12,41].

Based on these results, it is possible to identify a significant contribution of inoculants to the plant growth attributes of the investigated sesame cultivars, implying a beneficial action and a biosustainable alternative for the crop, with a view to reducing and/or replacing chemical fertilizers. The use of biofertilizers before, during, and after environmental stress can promote adjustments in plant defense mechanisms and increase soil water retention capacity, growth, and root performance [2,12,23,24,42]. 

The number of capsules per plant is a component of the final production, giving a cause-and-effect estimate: the more capsules per plant, the more seeds, which would also increase the seed mass. However, factors such as high temperature and water restriction can impair seed filling, as observed in this experiment. It is important to note that for the number of capsules per plant, strains pant001 and ESA 402 showed a positive and significant interaction for both cultivars (Figure 5a).

For the mass of a thousand seeds, the inoculants promoted higher averages than the nitrogen treatment, both in the irrigated condition and in the water deficit (Figure 5b). However, in the water deficit condition, the WN treatment presented the lowest average, being more sensitive to stress in the production of viable seeds for the two cultivars.

In the double interaction analysis (water regime x treatment) for the mass of a thousand seeds, the inoculants promoted higher averages than the nitrogen treatment both in the water deficit and in the irrigated conditions (Figure 6). These results suggest that biological inoculants can mitigate the effects of water deficit and, consequently, favor production since growth variables were also favored. Nitrogen fertilization was evaluated in sesame and found to be between 2.87 g and 3.8 g for the mass of one thousand seeds [43,44]. The values mentioned above correspond to those found in the present study, in which BRS Seda presented 3.42 g with ESA 402 and BRS Anahí 4.51 g with pant001 and ESA 402, both in the irrigated condition. It is worth noting that these findings were higher than the average values described in the literature for the two cultivars used in this study when grown under irrigation conditions close to field capacity.

In the analysis of the nitrogen content in the leaf tissue, an increase was observed in all treatments that had the water deficit condition for the two cultivars investigated (Figure 7). This accumulation of nitrogen in leaves in the final phase of the experiment may imply a deficiency in the reallocation of this nutrient since it is required for photosynthesis and gas exchange, in addition to fruit formation, influencing the behavior of plants under water deficit [45]. Studies carried out with nitrogen application at different phenological stages of sesame plants showed differences in nitrogen partitioning and remobilization [46]. These authors highlighted the importance of using a less soluble nitrogen source to increase the efficiency of its use during the crop cycle.

For the cultivar BRS Anahí, the treatments that were inoculated with the bacilli showed no significant difference when submitted to water deficit, despite that pant001, WN, and NN revealed the highest concentrations of nitrogen in leaves, which may imply a deficit in the allocation of the macronutrient to the formation of capsules and seeds. Still, in BRS Anahí, treatments submitted to water deficit obtained higher averages of up to 50% when compared to irrigated treatments. The BRS Seda cultivar also showed an increase in nitrogen concentration in treatments under water restriction. The highest concentration of nitrogen was in the nitrogen management itself; however, it was not statistically different from the treatments with the inoculants pant001, ESA 13, and ESA 402 (Figure 7).

### 2.3. Osmoregulation, Antioxidant Enzyme Complex and Chlorophyll Content

The quantification of total free proline in sesame leaves on the 8th day of water deficit showed a significant increase in all treatments with a significance level of *p* ≤ 0.01 for the triple interaction (Figure 8a). Proline plays an osmoregulatory role, binding to O_2_ and free radicals produced under water stress, so its action in inducing systemic resistance in plants becomes complex, attenuating the negative effects triggered by water deficit [47]. Considering the increase in the concentration of this osmoprotective solute in sesame plants in water deficit, it is possible to affirm that, despite the limiting condition imposed, the cultivars were able to synthesize this solute in a satisfactory way, observing the highest concentrations in the BRS Anahí in inoculated treatments and nitrogen management. Its synthesis also implies that the plants were truly in a condition of stress.

Osmoregulation in plants under low water potential depends on the synthesis and accumulation of osmoprotectants or osmolytes, such as soluble proteins, sugars and sugar alcohols, quaternary ammonium compounds, and amino acids, such as proline. The synthesis and accumulation of compatible cellular solutes help plants under water deficit conditions; this process is called osmotic adjustment [48]. Osmoprotectants are made up of various inorganic ions and organic solutes that act on the cellular osmotic potential and increase water use efficiency [49]. The accumulation of proline is already recognized as an important indicator of abiotic stress in plants, favoring intracellular homeostasis [50] and proving to be capable of increasing the capacity of plants to overcome lower water potentials, since this osmolyte has a particularity of buffering under the effect of water scarcity [51,52].

SOD increased its activity in treatments under water stress, implying an adjustment of the plants to the imposed condition (Figure 8b). However, BRS Anahí and BRS Seda associated with ESA 13 showed a decrease in SOD enzymatic activity. The highest concentration of SOD was observed in BRS Anahí in the NN treatment. In BRS Seda, the lowest concentration of SOD was in the association of the cultivar with ESA 402 and the highest concentration was in pant001, both in the stress condition. SOD, as an antioxidant enzyme, is considered the first line of defense for the plant cell. In the presence of ROS, the enzyme acts by dismuting O_2_^•−^ into H_2_O_2_, interfering with the concentration of ROS and the formation of ^•^OH radicals [53]. The accumulation of these free radicals damages the cellular arrangement, causing lipid peroxidation and cellular extravasation, in addition to affecting other biological molecules, including proteins and carbohydrates [54].

In the interaction between cultivars and inoculants, BRS Anahí with pant001, ESA 13, ESA 402, and WN showed higher CAT activity in the stress condition. For BRS Seda, interactions with pant001, ESA 441, and WN showed the highest CAT activities (Figure 8d). As can be seen, the performance of CAT varied; however, it is possible to state that some inoculants induced enzyme biosynthesis, helping to attenuate the oxidative effects triggered by water deficit and benefiting the cultivars. CAT is part of the antioxidant enzyme complex that defends plants that are subjected to abiotic stresses, such as water stress. Several sensitive, intermediate, and resistant sesame genotypes to water deficit presented CAT as one of the main enzymes that acts in the elimination of hydrogen peroxide (H_2_O_2_) generated in photorespiration and β-oxidation of fatty acids [55]. Under conditions of severe water stress, sesame inoculated with mycorrhiza (*Funneliformis mosseae* and *Rhizophagus irregularis*) presented an increase in catalase activity [56].

The highest activity of APX, under water deficit, was observed in the cultivar BRS Seda inoculated with pant001 and in BRS Anahí inoculated with ESA 13 (Figure 8c); BRS Anahí, when inoculated with ESA 441, showed increased APX activity in both water regimes. The functional activity of APX is complementary in plant defense; the biosynthetic inhibition observed in most treatments in this study may be linked to the increase in the activity of the first line of enzymatic defense, SOD (Figure 8b). Cowpea under water stress also showed a reduction in APX activity [57].

The antioxidant defense system includes a complex with several antioxidant enzymes such as SOD, CAT, and APX. When plants undergo oxidative stress due to some factor related to adverse environmental conditions, such as water or saline stress, these enzymes act in several subcellular sections to prevent damage to the plant cell [58,59,60,61]. Antioxidant enzymes act in a concatenated manner to establish maximum efficiency in plant defense under any adverse condition. Catalases are the main enzymes that convert hydrogen peroxide resulting from photorespiration into H_2_O and molecular oxygen (O_2_) [62]. As well as catalases, ascorbate peroxidase also acts in the primary defense, attenuating the deleterious effects of ROS in the plant cell, especially in plants under water deficit [57]. SOD enzymes catalyze the dismutation of the superoxide radical into H_2_O_2_ + O_2_, CAT, and APX, which can break down H_2_O_2_ → H_2_O + O_2_, antioxidants responsible for defending against free radicals, promoting detoxification caused by ROS that cause damage and compromise the plant cell functions [63].

In plants such as sesame, stress induces a complex plant response that depends on several factors, such as duration of stress, phenological phase, soil type, and genotype [64]. Sesame bears the physiological mark of stomatal resistance, which in turn provides an extension of its tolerance to drought [65] and in concomitant response to these processes are biochemical reactions that potentiate the defense system, generating a joint protection network that aims to maintain the plant’s survival in the face of the limited conditions of its natural activities.

The sesame cultivars, despite the water restriction, showed positive interaction with the inoculants, increasing the concentration of chlorophyll a in the treatments with ESA 402 for BRS Anahí and ESA 13 and ESA 402 for BRS Seda (Figure 9a). In the evaluation of chlorophyll b, an increase was observed in the association of BRS Seda with ESA 13 (Figure 9b), and for total chlorophyll, BRS Seda with ESA 13 and ESA 402 was more efficient in the biosynthesis of pigments, presenting values higher than the control (Figure 9c). As for the carotenoid content, BRS Anahí inoculated with ESA 402 and BRS Seda with ESA 13, ESA 402, and ESA 441 promoted an increase superior to the control (Figure 9d). A decrease in chlorophyll a and b levels with increasing water stress was observed in sesame, with a reduction of 60% and 26%, respectively, compared to plants under optimal irrigation [56]. On the other hand, the authors highlighted an increase in carotenoids in genotypes under water deficit inoculated with mycorrhizal fungi. 

The WN treatments showed higher values than the control for chlorophyll a, b, total, and carotenoids in BRS Seda and for the carotenoid variable in BRS Anahí. This behavior may be linked to the fact that nitrogen is an essential component of the chemical structure of pigments, so fertilization with this nutrient enables the synthesis of photosynthetic compounds. These pigments are directly linked to the nutritional status of plants, especially nitrogen, given that the total content of this nutrient in the leaf ratio is concentrated in chloroplasts [66]. Water stress usually results in the destruction of chloroplasts and consequently in a decrease in chlorophyll, in addition to the activity of enzymes in the Calvin cycle during the process of photosynthesis [67].

The results of this work are promising in relation to the proposed objectives and can be used as a basis to guide other field studies, knowing in advance the behavior of sesame genotypes (BRS Anahí and BRS Seda) in relation to inoculation with bacilli under water restriction. The summary of the main physiological and biochemical variables is described in Table 1.

## 3. Materials and Methods

### 3.1. Cultivation of Bacteria and Preparation of Sesame Seeds

The *Bacillus* spp. strains ESA 13 [6], ESA 402 [7,40], and ESA 441 [68] were obtained from the “Coleção de Culturas de Micro-organismos de Interesse Agrícola da Embrapa Semiárido” (Embrapa Semiárido, Petrolina-PE, Brazil). They were streaked in LB solid medium (Luria Bertani) and incubated for 24 h at 28 °C. They were then subcultured in liquid LB medium and incubated at 28 °C, 180 rpm, for 72 h, until the exponential phase of bacterial growth (1.0 × 10^9^ CFU mL^−1^) [69]. The inoculant containing the strain pant001 (*Bacillus subtilis*—Panta Premium) [70] was provided by the Geoclean company, and, together with the bacteria subcultured in liquid medium, they were used directly in the inoculation of the seeds.

Sesame seeds were disinfected with pure ethanol for 15 s, 1% sodium hypochlorite for 1 min, and finally washed 10 times with sterile distilled water [69]. Then, the sesame seeds were soaked in the inoculants for 10 min and sown in pots.

### 3.2. Implementation and Conduction of the Experiment in a Greenhouse

The experiment was carried out in a greenhouse at Embrapa Algodão, Campina Grande-PB, Brazil (07°13′ S; 53°31′ W) [71]. Two sesame cultivars (BRS Seda and BRS Anahí) were grown in pots with a capacity of 20 L; the pots were filled with sandy loam soil. The soil was previously analyzed at the Laboratory of Soils and Plant Nutrition of Embrapa Algodão [72] and corrected with dolomitic limestone. The substrate, according to the treatment, was fertilized with nitrogen (ammonium sulfate, 95 kg ha^−1^), distributed in two applications, the first at 10 days after emergence and the second at the beginning of flowering. Phosphorus (single superphosphate) and potassium (KCl) (110 kg ha^−1^ and 34 kg ha^−1^, respectively) were applied for all treatments, based on the analysis of the soil [73].

Irrigation was performed daily, trying to maintain soil moisture close to field capacity and suspended in stressed treatments from the 30th day after emergence (beginning of flowering), for eight days, and then rehydrated.

The experimental design was completely randomized, established by a random process in a factorial scheme: 2 (cultivars) × 2 (water regime) × 6 (inoculation/fertilization treatments), totaling 24 treatments with 5 replications. The treatments were characterized as: (i) nitrogen management (with N, WN) (ammonium sulfate, 21% N); (ii) absolute control, without nitrogen (no N, NN); (iii) management with 4 inoculants based on bacilli, strains pant001, ESA 13, ESA 402, and ESA 441; and with and without irrigation.

### 3.3. Physiological Measures

The sesame plants were evaluated in the morning, between 9:00 a.m. and 11:00 a.m., during the water deficit period (8 days), using a portable photosynthesis analyzer (IRGA—Infra Red Gas Analyzer, model LCpro-SD), without an artificial carbon source and with an artificial light source of 1200 µmol m^−2^ m^−1^. The following parameters were evaluated: stomatal conductance (gs) (mol m^−2^ m^−1^); photosynthesis (A) (μmol m^−2^ m^−1^); transpiration (E) (mmol m^−2^ m^−1^) and internal concentration of CO_2_ (Ci) (μmol mol^−1^) [74,75]. From the obtained data, the instantaneous efficiency of carboxylation (iEC) between A and Ci (A/Ci) and the instantaneous transpiration efficiency (iTE), calculated as the photosynthesis/transpiration ratio, between A and E (A/E) [75] were estimated. To monitor stomatal closure, three assessments were performed during the water suspension period (1D; 4D; 8D—“D = Days”). When the plants reached approximately 90% stomatal closure, they were rehydrated (criterion adopted by the research team). The fluorescence of the plants was also evaluated [76], using a portable fluorometer, with which the initial fluorescence (Fo), variable fluorescence (Fv), mean fluorescence (Fm), and the relationship between Fv/Fm were estimated.

### 3.4. Nitrogen Content of the Shoot

At the end of the crop cycle, a sample of the aerial part of the plants was collected, stored in kraft paper bags, and placed in an oven with forced air circulation at 65 °C for 72 h, then ground in a mill. The nitrogen analysis of the aerial part of the plants was based on the sulfuric digestion method developed by Kjeldahl [77]. From the nitrogen content, the total nitrogen accumulated in the shoot was calculated by multiplying the nitrogen content by the dry mass of the shoot [78].

### 3.5. Antioxidant Activities and Proline Content

A sample of fresh leaves was collected on the eighth day of water restriction, immediately immersed in liquid N_2_, and then stored at −80 °C. For protein extraction, the leaves (200 mg) were macerated in liquid N_2_ and 3 mL of 0.1 M potassium phosphate buffer, pH 7.0, containing 100 mM EDTA, 1 mM L-ascorbic acid, and 4% polyvinylpolypyrrolidone (PVP) were added. The extracts were centrifuged at 12000 rpm for 10 min at 4 °C and the supernatants were transferred to new microtubes. Protein quantification was determined by Bradford’s method [79] in a spectrophotometer at 595 nm. Superoxide dismutase (SOD) activity was determined and analyzed in a spectrophotometer at 560 nm [80]. The results were expressed in UA g MF^−1^ activity (Activity Unit g Fresh Pasta). Catalase activity (CAT) was determined, and the reading was performed in a spectrophotometer at 240 nm [81]. Ascorbate peroxidase (APX) activity was determined and analyzed in a spectrophotometer at 290 nm [82]. Free proline content was determined and analyzed in a spectrophotometer at 520 nm [83].

### 3.6. Chlorophyll and Carotenoid Content

Chlorophyll content of a, b, and total (a + b), and carotenoid contents were determined using the 80% acetone extraction method [84]. The entire procedure was performed in the presence of green light, thus avoiding the degradation of chlorophyll. In this methodology, 200 mg of leaves was macerated in liquid N_2_ and then solubilized in 10 mL of 80% acetone. Subsequently, the solution was filtered through qualitative filter paper and read at the following absorbances: 470, 646.8, 663.2, and 710 nm.

### 3.7. Biomass and Growth Measures 

At the end of the crop cycle, the following growth characteristics were evaluated: plant height (cm), measured from the soil surface to the apex of the main stem, using a metric tape; stem diameter, measured with a caliper; number of capsules per plant; dry mass of shoots (g) and roots (g), determined by drying the material in an oven with forced air circulation at 65 °C, for approximately 72 h, until reaching a constant mass, and weighing on a precision scale; and mass of 1000 seeds [85].

### 3.8. Statistical Analysis

The collected data were analyzed using the SISVAR software version 5.6 [86], submitted to an analysis of variance (*p* ≤ 0.05), and the means compared by Tukey’s test (*p* ≤ 0.05). They were also submitted to the Shapiro–Wilk normality test to verify and correct data heterogeneity.

## 4. Conclusions

The cultivars BRS Anahí and BRS Seda showed semi-tolerant characteristics to water deficit. It was possible to observe a morphophysiological and biochemical adjustment, with inoculation with *Bacillus* spp. being a relevant factor for the results obtained. Under water deficit, the bacterial strains promoted positive effects on the plant height and weight of one thousand seeds variables. The interactions of BRS Anahí x ESA 13 and BRS Seda x ESA 402 promoted the greatest increases in weight of one thousand seeds with 42% and 34%, respectively. In the irrigated condition, it was observed that all inoculants promoted an increase of up to 34% for the weight of one thousand seeds in both cultivars.

In this perspective, the biofertilizer containing the strains assessed in the present study (mainly ESA 13 and ESA 402) should constitute an important agricultural input capable of mitigating the effects of the water deficit on the plants and providing positive effects on the increase in sesame production, providing greater economic viability to the crop.

## Figures and Tables

**Figure 1 plants-12-01337-f001:**
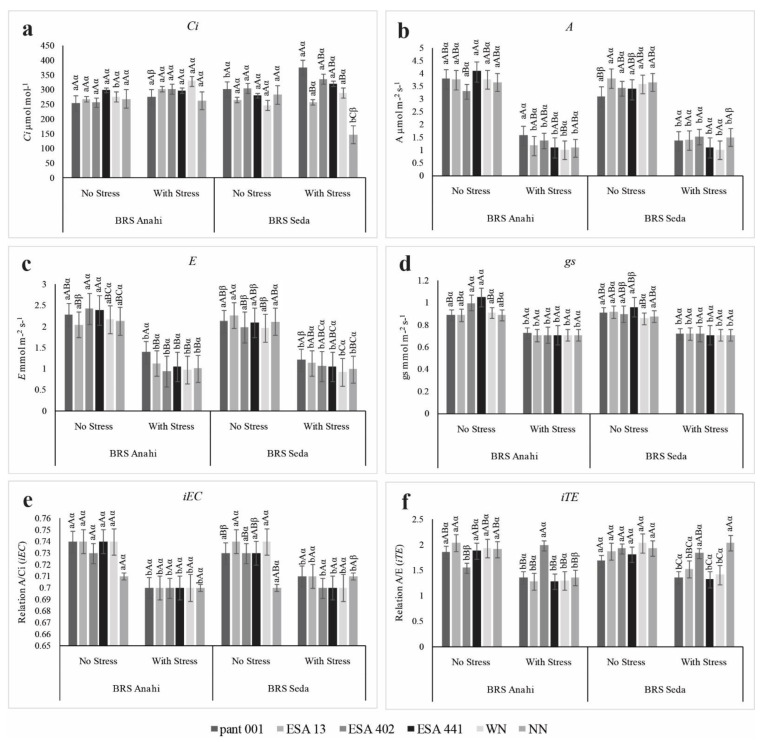
(**a**) Internal CO_2_ concentration (Ci), (**b**) photosynthesis (A), (**c**) transpiration (E); (**d**) stomatic conductance (gs), (**e**) instantaneous efficiency of carboxylation (iEC), (**f**) instantaneous transpiration efficiency (iTE) in sesame cultivars (BRS Anahí and BRS Seda) inoculated with bacilli, under water deficit and different sources of nitrogen. WN-with nitrogen and NN-no nitrogen. Triple interactions: lowercase letters compare the water regime within each cultivar; capital letters compare treatments with inoculants within each cultivar; and Greek letters compare the cultivars.

**Figure 2 plants-12-01337-f002:**
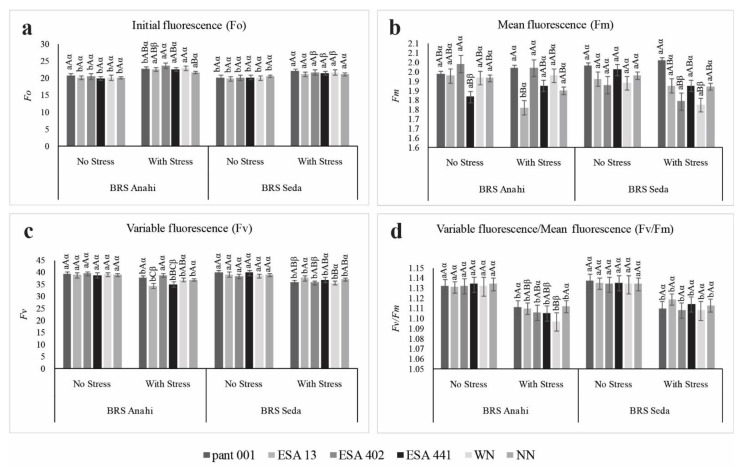
(**a**) Initial fluorescence (Fo), (**b**) average fluorescence (Fm), (**c**) variable fluorescence (Fv); (**d**) relationship between Fv/Fm, in sesame cultivars (BRS Anahí and BRS Seda) inoculated with bacilli, under water deficit and different nitrogen sources. WN-with nitrogen and NN-no nitrogen. Triple interactions: lowercase letters compare the water regime within each cultivar; capital letters compare treatments with inoculants within each cultivar; and Greek letters compare the cultivars.

**Figure 3 plants-12-01337-f003:**
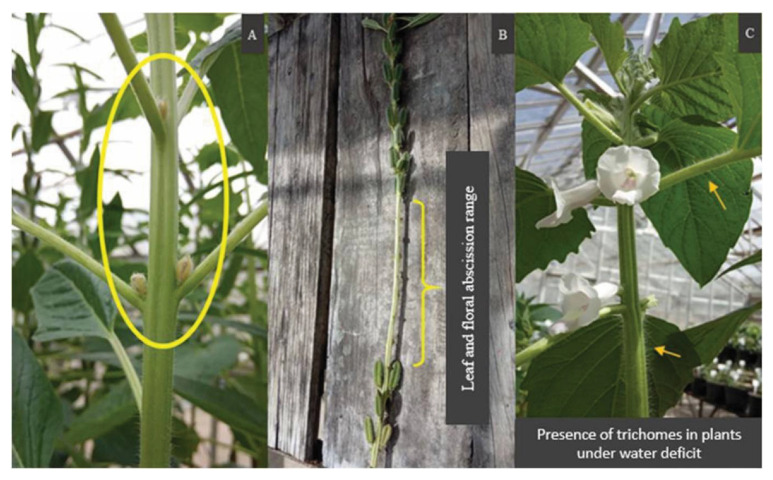
Observation of the events caused by water deficit in plants at BRS Anahí: (**A**) floral abortion; (**B**) leaf and floral abortion interval; and (**C**) presence of trichomes.

**Figure 4 plants-12-01337-f004:**
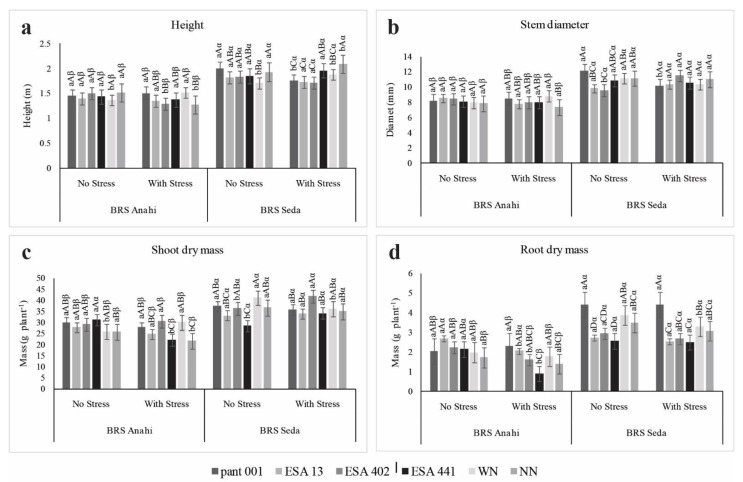
Growth characteristics of sesame cultivars (BRS Anahí and BRS Seda) inoculated with bacilli, under water deficit and different nitrogen sources. WN-with nitrogen and NN-no nitrogen. Plant height (**a**), stem diameter (**b**), shoot dry mass (**c**) and root dry mass (**d**). Lowercase letters compare the water regime within each cultivar; capital letters compare treatments within each cultivar; Greek letters compare cultivars.

**Figure 5 plants-12-01337-f005:**
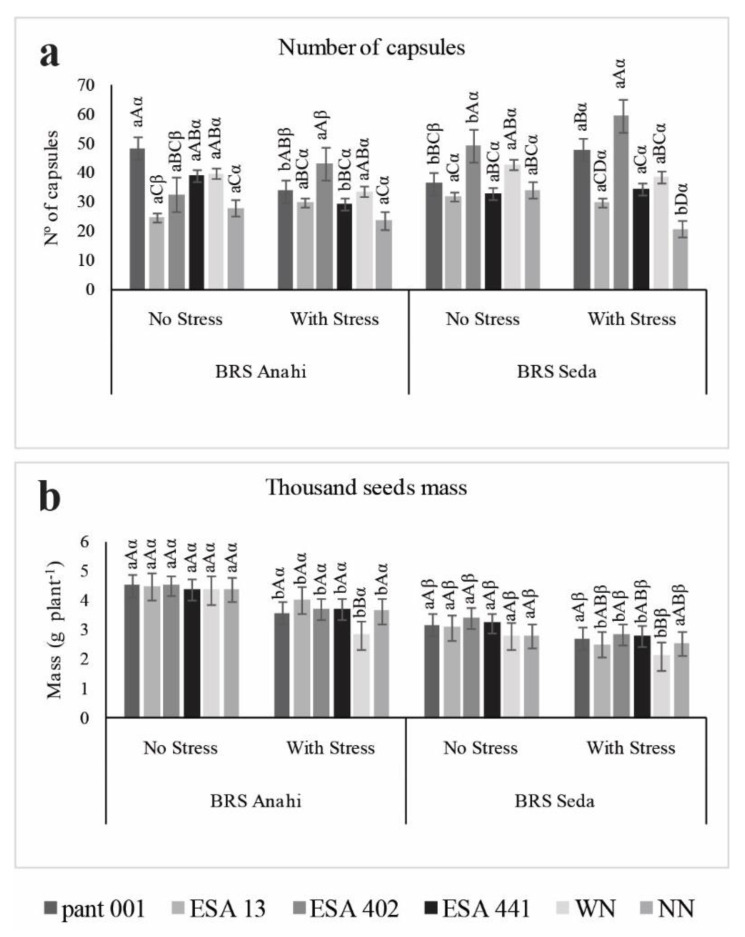
Number of capsules per plant (**a**) and weight of one thousand seeds (**b**) of two sesame cultivars (BRS Seda and BRS Anahí) inoculated with bacilli and under water restriction. WN-with nitrogen and NN-no nitrogen. Lowercase letters compare the water regime within each cultivar; capital letters compare treatments within each cultivar; Greek letters compare cultivars, according to Tukey’s test at 5% probability.

**Figure 6 plants-12-01337-f006:**
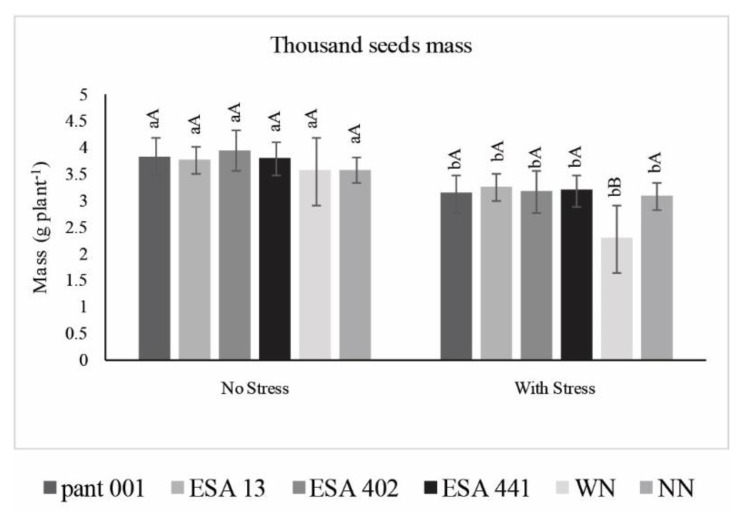
Interaction between water regime x treatments, with inoculants based on bacilli, with significance at *p* ≤ 0.05 Tukey test for the mass of one thousand seeds of sesame cultivars (BRS Seda and BRS Anahí), under water restriction. Water regimen: no stress and with stress. WN-with nitrogen and NN-no nitrogen. Lowercase letters compare water regime; capital letters compare treatments within the water regime.

**Figure 7 plants-12-01337-f007:**
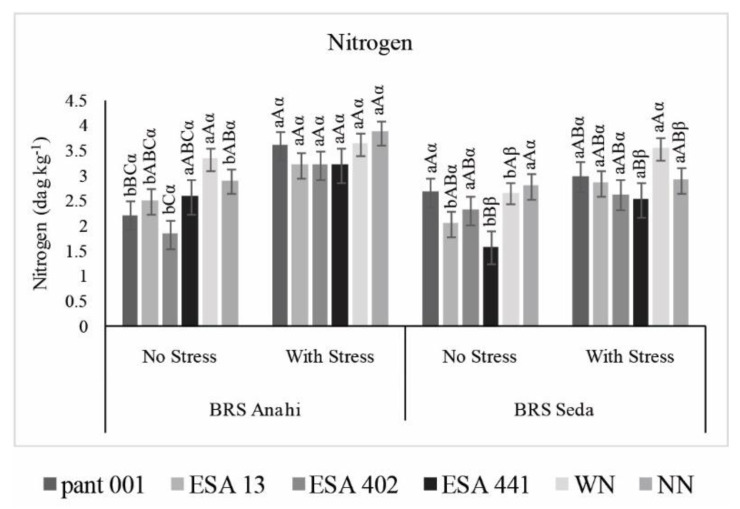
Accumulation of nitrogen in leaves of two sesame cultivars (BRS Seda and BRS Anahí) inoculated with bacilli, under water restriction, from crude protein (*p* ≤ 0.05). WN-with nitrogen and NN-no nitrogen. Lowercase letters compare the water regime within each cultivar; capital letters compare treatments within each cultivar; Greek letters compare cultivars.

**Figure 8 plants-12-01337-f008:**
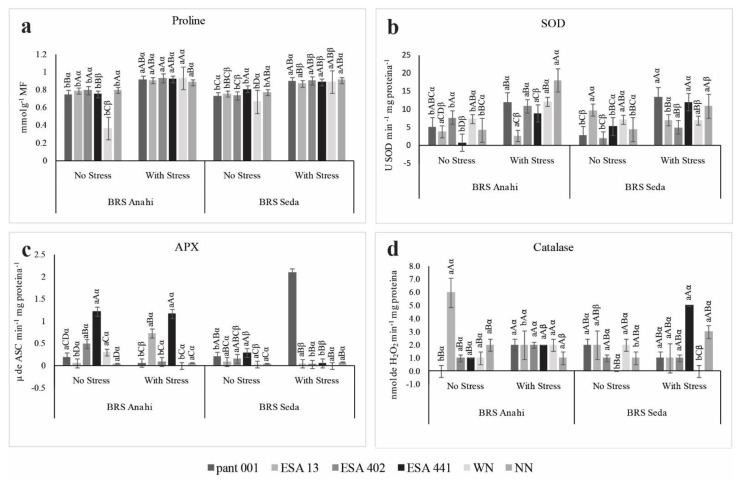
Concentration of proline (**a**), superoxide dismutase (SOD) (**b**), ascorbate peroxidase (APX) (**c**) and catalase (CAT) (**d**) as a function of water deficit in two sesame cultivars (BRS Seda and BRS Anahí) inoculated with bacilli. WN-with nitrogen and NN-no nitrogen.

**Figure 9 plants-12-01337-f009:**
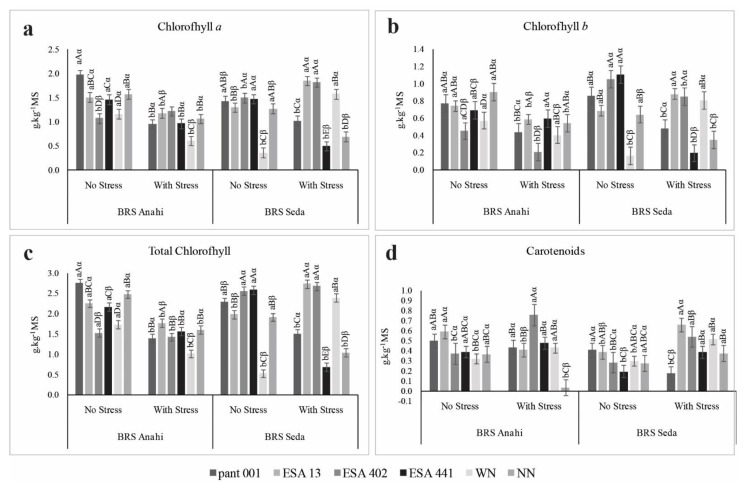
Photosynthetic pigments: chlorophyll a (**a**), chlorophyll b (**b**), total chlorophyll (**c**), and carotenoid (**d**) in sesame plants (BRS Seda and BRS Anahí) inoculated with bacilli, under water deficit. WN-with nitrogen and NN-no nitrogen.

**Table 1 plants-12-01337-t001:** Summary of the main physiological and biochemical variables analyzed in the interaction between the two sesame genotypes (BRS Anahí and BRS Seda) and the bacilli-based inoculants, submitted to water restriction and compared to the control (*p* ≤ 0.01 and *p* ≤ 0.05 by the Tukey test).

	Ci	E	A	gs	Proline	SOD	APX	CAT
BRS Anahí								
pant 001	˄	**˅**	**˅**	**˅**	˄	˄	**˅**	˄
ESA 13	˄	**˅**	**˅**	**˅**	˄	**˅**	˄	**˅**
ESA 402	˄	**˅**	**˅**	**˅**	˄	˄	**˅**	˄
ESA 441	˄	**˅**	**˅**	**˅**	˄	˄	**˅**	˄
Nitrogen	˄	**˅**	**˅**	**˅**	˄	˄	**˅**	˄
No Nitrogen	**˅**	**˅**	**˅**	**˅**	˄	˄	˄	**˅**
BRS Seda								
pant 001	˄	**˅**	**˅**	**˅**	˄	˄	˄	**˅**
ESA 13	**˅**	**˅**	**˅**	**˅**	˄	**˅**	**˅**	**˅**
ESA 402	˄	**˅**	**˅**	**˅**	˄	˄	**˅**	˄
ESA 441	˄	**˅**	**˅**	**˅**	˄	˄	**˅**	˄
Nitrogen	˄	**˅**	**˅**	**˅**	˄	**˅**	**˅**	**˅**
No Nitrogen	**˅**	**˅**	**˅**	**˅**	˄	˄	˄	˄

Ci—internal concentration of CO_2_; E—transpiration; A—photosynthesis; gs—stomatal conductance; SOD—superoxide dismutase; APX—ascorbate peroxidase; CAT—catalase; ˄—increased content; **˅**—reduced content.

## Data Availability

All other data are presented in the paper.

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
