# Peer review of "Bacilli Rhizobacteria as Biostimulants of Growth and Production of Sesame Cultivars under Water Deficit"

_plants, 2023, doi:10.3390/plants12061337_

Round 1
Reviewer 1 Report
In the present manuscript, the authors describe the adverse effect of drought stress on sesame plant and shoes that plant-growth-promoting bacteria are a valuable source to overcome the harmful effect of drought. This Manuscript adds valuable information to the existing knowledge of biofertilizers development against drought and could be a useful resource to develop a novel strategy to mitigate drought stress. This study could attract wide research in the field. However, before being published, I have a few minor comments for strengthening.
None of the provided keywords are frequently used in the manuscript, and I could not find a few in the main text; consider providing relative keywords.
Lines 56 and 57 do not make any sense; revise.
Can the authors provide error bars in each of the graphs of the manuscript?
Line 402 "streaked" spelling error; correct it.
Lines 409 and 410 explain correctly.
Section 3.2, did the authors sterilize potting soil? If not, then they can justify the effect because of inoculated formulation and not soil microbiome.
Below are the links for relative references; consider citing them in the Introduction and discussion.
https://link.springer.com/article/10.1007/s10725-014-0008-8
https://link.springer.com/article/10.1007/s11104-015-2436-2
https://academic.oup.com/jambio/article/131/3/1417/6715670
Author Response
Dear reviewer,
We appreciate the valuable contributions to improve our manuscript. The suggestions were accepted. The answers to each question are described below.
None of the provided keywords are frequently used in the manuscript, and I could not find a few in the main text; consider providing relative keywords.
Answer - We changed the keywords in order to meet the suggestion: abiotic stress; water restriction; bacilli; osmoprotection system; leaf gas exchange.
Lines 56 and 57 do not make any sense; revise.
Answer - The text has been revised and improved: “Sesame is used in the production of various foods, in the medicinal and pharmaceutical industry, also consumed in natura and in animal feed”.
Can the authors provide error bars in each of the graphs of the manuscript?
Answer - Error bars have been inserted into the graphs.
Line 402 "streaked" spelling error; correct it.
Answer - The word ‘striated’ has been replaced by ‘striaked’.
Lines 409 and 410 explain correctly.
Section 3.2, did the authors sterilize potting soil? If not, then they can justify the effect because of inoculated formulation and not soil microbiome.
Answer - The soil was not sterilized, however, it was the same soil used in the control treatments. The controls served as a comparison parameter.
Below are the links for relative references; consider citing them in the Introduction and discussion.
https://link.springer.com/article/10.1007/s10725-014-0008-8
https://link.springer.com/article/10.1007/s11104-015-2436-2
https://academic.oup.com/jambio/article/131/3/1417/6715670
Answer - We appreciate the references suggestions, they were introduced in the manuscript, as well as several others.

Reviewer 2 Report
The manuscript entitled “Rhizobacteria as biostimulants of growth and production of 2 sesame cultivars under water deficit” submitted for possible publication to “plants” is an interesting review study. The manuscript covers the contents to some extent positively, but there are some key points lacking. The manuscript can be accepted after major revision as per comments given below:
Title
Ø The keywords from title are not included, keywords from title should be added.
Abstract
Ø The aim of the study is missing, aim of the study should be added in abstract.
Ø The results described are very short, some more words about results should be added
Ø Research gap is missing
Introduction
Ø Update the introduction with latest references preferably among 2020-2023, reference before 10 years of this study should try to be avoided until its necessary to include some old reference.
Ø References should be cross referenced for reviewers ease.
Ø The studies about bacilli-based inoculants are missing, some latest studies should be added
Results and discussion
Ø The results are briefed and well elaborated, the authors are appreciated for their hard work.
Ø The size of images and font size of lettering on the graphs is very low, for ease in reading it should be increased.
Material and methods
Ø The Implementation and conduction of the experiment in a greenhouse, citation required.
Ø Physiological measures, citation required.
Ø Biomass and growth measures, citation required.
Conclusion
Ø The conclusion is very short, some more description of results in conclusion can make manuscript more attractive for reviewers.
Ø Research gap and future recommendations are missing. It is suggested that to add research gap and future recommendations.
Author Response
Dear reviewer,
We appreciate the valuable contributions to improve our manuscript. The suggestions were accepted. The answers to each question are described below.
Title
Ø The keywords from title are not included, keywords from title should be added.
Answer - We changed the keywords in order to meet the suggestion: abiotic stress; water restriction; bacilli; osmoprotection system; leaf gas exchange.
Abstract
Ø The aim of the study is missing, aim of the study should be added in abstract.
Answer - The suggestion was accepted.
Ø The results described are very short, some more words about results should be added
Answer - The abstract has 199 words, within the limit acceptable by Plants (200 words maximum). However, we emphasize that all variables were presented in the results in a summarized form "A positive effect of inoculants was observed for all characteristics evaluated, contributing to improvement in plant physiology (refers to gas exchange), induction of biochemical responses (refers to antioxidant activities, proline, nitrogen, chlorophyll and carotenoid) vegetative development, and productivity (refers to biomass and growth measures)”.
Ø Research gap is missing
Answer - Sorry, we have not identified the missing research gap.
Introduction
Ø Update the introduction with latest references preferably among 2020-2023, reference before 10 years of this study should try to be avoided until its necessary to include some old reference.
Ø References should be cross referenced for reviewers ease.
Ø The studies about bacilli-based inoculants are missing, some latest studies should be added.
Answer - Several references have been inserted in the Introduction to meet the suggestions.
Results and discussion
Ø The results are briefed and well elaborated, the authors are appreciated for their hard work.
Answer - Thank you.
Ø The size of images and font size of lettering on the graphs is very low, for ease in reading it should be increased.
Answer - The figures were adjusted to meet the suggestions.
Material and methods
Ø The Implementation and conduction of the experiment in a greenhouse, citation required.
Ø Physiological measures, citation required.
Ø Biomass and growth measures, citation required.
Answer - All required citations have been added.
Conclusion
Ø The conclusion is very short, some more description of results in conclusion can make manuscript more attractive for reviewers.
Ø Research gap and future recommendations are missing. It is suggested that to add research gap and future recommendations.
Answer - We have improved the conclusions to meet the suggestions.
